# A Comparison of Laboratory Simulation Methods of Iron Contamination for FCC Catalysts

**Yitao Liao [1,2], Tao Liu [2], Huihui Zhao [2] and Xionghou Gao [2,\*]**

[1] College of Chemistry and Chemical Engineering, Northwest Normal University, Lanzhou 730070, China; liaoyitao@petrochina.com.cn

[2] Lanzhou Petrochemical Research Center, Petrochemical Research Institute, PetroChina, Lanzhou 730060, China; liutao5@petrochina.com.cn (T.L.); zhaohuihui3@petrochina.com.cn (H.Z.)

\* Correspondence: gaoxionghou@petrochina.com.cn; Tel.: +86-010-8016-5606

**Abstract:** Two different methods of simulating iron contamination in a laboratory were studied. The catalysts were characterized using X-ray diffraction, N$_2$ adsorption–desorption, and SEM-EDS. The catalyst performance was evaluated using an advanced cracking evaluation device. It was found that iron was evenly distributed in the catalyst prepared using the Mitchell impregnation method and no obvious iron nodules were found on the surface of the catalyst. Iron on the impregnated catalyst led to a strong dehydrogenation capacity and a slight decrease in the conversion and bottoms selectivity. The studies also showed that iron was mainly in the range of 1–5 μm from the edge of the catalyst prepared using the cycle deactivation method. Iron nodules could be easily observed on the surface of the catalyst. The retention of the surface structure in the alumina-rich areas and the collapse of the surface structure in the silica-rich areas resulted in a continuous nodule morphology, which was similar to the highly iron-contaminated equilibrium catalyst. Iron nodules on the cyclic-deactivated catalyst led to a significant decrease in conversion, an extremely high bottoms yield, and a small increase in the dehydrogenation capacity. The nodules and distribution of iron on the equilibrium catalyst could be better simulated by using the cyclic deactivation method.

**Keywords:** Mitchell impregnation; cyclic deactivation; iron contamination; iron nodules

## 1. Introduction

How to test a fluid catalytic cracking (FCC) catalyst's performance and metal tolerance in the laboratory and calibrate the results to simulate the industrial performance is an ongoing discussion. Another important aspect of catalyst testing is the preparation of the catalyst [1]. In evaluating a fluid cracking catalyst's performance in lab-scale testing, selecting the proper catalyst deactivation method is just as important as the testing itself [2].

When heavy metals are deposited on the catalyst, the physical and chemical properties of the catalyst will be affected by the distribution, valence state, and existing forms of the heavy metals. Therefore, it is difficult to accurately simulate the industrial heavy-metal-polluted equilibrium catalyst by selecting the laboratory deactivation protocols. Recently, due to the relatively low supply of light and low-sulfur crude oil, which has a relatively high market price, refiners are obliged to process increasing quantities of heavy crude by cracking them into lighter distillates [3,4]. With the increasing trend of heavy and poor-quality FCC feedstocks, the quantity and species of heavy metal deposition, such as Ni [5,6], V [7,8], Ca [9,10], Na [11], and especially iron [12], on the catalyst have gradually increased. Iron deactivation effects can be subdivided into the following two classifications: (1) iron nodules may affect the catalyst's apparent bulk density (ABD) and fluidization [13] and (2) increasing the amount of iron will cause a loss of activity and bottoms cracking, as well as increased SOx emissions and coke on the regenerated catalyst (CRC) in partial burn units [14].

Multiple kinds of heavy metals deposition on a catalyst increase the difficulty of selecting a deactivation protocol. The detrimental effects of nickel and vanadium that were studied using deactivation protocol simulations have been described elsewhere. Buurmans et al. [15] investigated the effect of nickel and vanadium deactivation on the structure and acidity of FCC catalyst particles by using steaming, cyclic deactivation, and the Mitchell impregnation method. Nguyen et al. [16] investigated the effect of hydrothermal conditions on the catalytic deactivation of an FCC catalyst by using a cyclic propylene steaming (CPS) method. Etim et al. [17,18] investigated the role of nickel on a vanadium-poisoned FCC catalyst and the effect of vanadium contamination on the framework and micropore structure by using the Mitchell impregnation method. However, few studies on the simulation and catalytic performance of iron contamination have been reported. Nobody has successfully simulated the iron nodules observed on the surface of an iron-contaminated equilibrium catalyst in the laboratory. If there is no accurate method for simulating iron deactivation, the research on the impact of iron contamination and the development of iron tolerance technology in the laboratory will not have real and reliable theoretical support, and sometimes wrong conclusions may be obtained. Therefore, in order to understand the real effect of iron on the catalyst, it is necessary to establish an accurate and effective simulation method of iron deactivation.

Laboratory catalyst deactivation protocols are used in order to simulate long-term commercial catalyst deactivation in an accelerated way [19]. A traditional way was proposed using the Mitchell impregnation method, which involves the impregnation of the catalyst with metal solutions, followed by calcination and a steam treatment [20]. An advanced method was to use the cyclic deactivation (CD) method, which has been described elsewhere. By using the CD method, Psarras et al. [21] investigated the accessibility effect on the irreversible deactivation of FCC catalysts. Almas et al. [22] studied the transformations of FCC catalysts and carbonaceous deposits during repeated reaction–regeneration cycles. Rainer et al. [23] investigated the Akzo Accessibility Index (AAI) of iron-contaminated catalysts under different conditions. One CD section consisted of repeated cycles of cracking, stripping, and regeneration treatment. The deactivation of the FCC catalyst in a CD unit is a simulation of the deactivation processes in a commercial plant.

In this work, a commercial catalyst (CAT-BASE) was characterized and tested in an advanced cracking evaluation (ACE) unit following cyclic deactivation (CAT-CD) and Mitchell impregnation with different iron species (CAT-MM1 using iron chloride and CAT-MM2 using iron naphthenate) in order to obtain a more direct comparison of the two laboratory methods regarding an iron-contaminated equilibrium catalyst simulation. Finally, the nodules, iron distribution, and the performance of the iron contaminated equilibrium catalyst could be better simulated by using the cyclic deactivation method.

## 2. Results

### 2.1. Iron Deposited on the Catalyst Using Different Methods

In order to study the effect of different contamination methods on the properties and catalytic performance of an iron-contaminated catalyst, three catalysts based on CAT-BASE were prepared by using iron chloride and iron naphthenate as precursors. An iron-contaminated equilibrium catalyst (E-cat) was selected as the target catalyst for different deactivation methods. CAT-BASE and E-cat were collected from the industrial plant of Dalian West Pacific Petrochemical Co., Ltd. (WEPEC, Dalian, China). The metal contents of the catalysts were obtained using XRF. The properties of the catalysts are shown in Table 1.

There was a certain amount of iron (1828 µg/g) on CAT-BASE, which came from the kaolin used to prepare the catalyst but it had no poisoning effect on the performance of the catalyst [24]. Three different metal-depositing methods were used to load almost the same amount of iron onto the catalyst. Finally, the target iron deposition was 12,000 µg/g, where the actual iron deposition on CAT-MM1 was 11,480 µg/g and that on CAT-MM2 was 11,440 µg/g. A similar amount of iron (11,690 µg/g) was deposited onto CAT-CD.

**Table 1.** Properties of different catalysts.

| Catalysts | $\omega$ (Na$_2$O) (%) | $\omega$ (Al$_2$O$_3$) (%) | S.A. (m$^2\cdot$g$^{-1}$) | P.V.$_{total}$ (cm$^3\cdot$g$^{-1}$) | Unit Cell Size (nm) | Fe on Catalyst ($\mu$g$\cdot$g$^{-1}$) |
|---|---|---|---|---|---|---|
| CAT-BASE | 0.16 | 48.6 | 269 | 0.42 | 2.462 | 1828 |
| CAT-MM1 | 0.16 | 48.6 | 149.4 | 0.21 | 2.440 | 13,308 |
| CAT-MM2 | 0.16 | 48.6 | 137.6 | 0.20 | 2.439 | 13,268 |
| CAT-CD | 0.16 | 48.6 | 122.5 | 0.16 | 2.436 | 13,518 |
| E-cat | 0.14 | 47.4 | 109.7 | 0.15 | 2.437 | 10,471 |

S.A.: Surface Area; P.V.: Pore Volume; CAT-BASE: commercial catalyst, CAT-MM1: catalyst after using iron chloride, CAT-MM2: catalyst after using iron naphthenate, CAT-CD: catalyst after cyclic deactivation, E-cat: iron-contaminated equilibrium catalyst.

## 2.2. Iron-Contaminated Catalysts' Morphology

The surface morphology of an industrial iron-contaminated equilibrium catalyst is described elsewhere [25–27]. The iron nodules can be clearly observed on the E-cat surface in Figure 1a. During the formation of iron nodules, the iron formed a eutectic and released a lot of heat to melt the silica-alumina matrix layer [28]. A dense layer was formed that is typically a few microns in depth, as shown in Figure 2a.

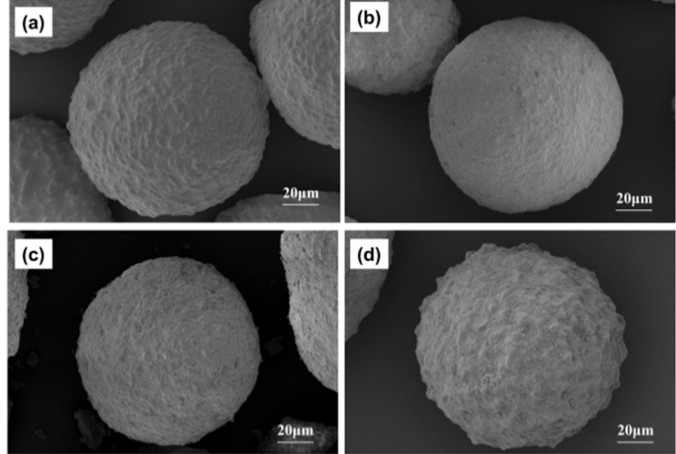

**Figure 1.** SEM micrographs of (**a**) E-cat, (**b**) CAT-MM1, (**c**) CAT-MM2, and (**d**) CAT-CD.

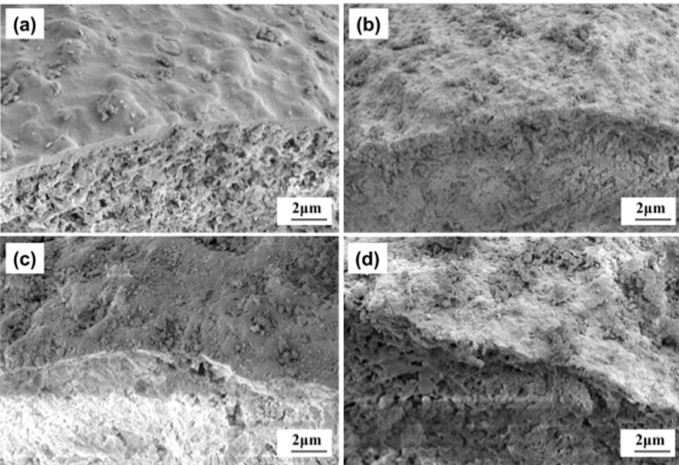

**Figure 2.** SEM micrographs of the catalyst cross-section: (**a**) E-cat, (**b**) CAT-MM1, (**c**) CAT-MM2, and (**d**) CAT-CD.

The surfaces of CAT-MM1 and CAT-MM2 in Figure 1b,c were relatively smooth. The contamination sources used for the impregnation of CAT-MM1 and CAT-MM2 were iron chloride and iron naphthenate, where their lengths [25] were less than the pore diameter of CAT-BASE. During the impregnation process, iron chloride and iron naphthenate may have entered into the pore of the catalyst; therefore, no obvious iron nodules and boundary can be observed on the catalyst surface and cross-section (Figure 2b,c)

Iron nodules similar to those in Figure 1a can be observed on the surface of CAT-CD in Figure 1d. A few microns of the layer can also be observed in Figure 2d. Therefore, compared with the Mitchell impregnation method, the cyclic deactivation method could truly simulate the surface morphology of an industrial iron-contaminated equilibrium catalyst.

*2.3. Iron Distribution on the Catalyst*

Within the industrial FCC unit, the metals are usually deposited on the equilibrium catalyst with different deposition profiles. For example, nickel is known to deposit on the outer part of the catalyst [29]. Vanadium mainly diffuses from the outside to the inside [30,31]. Iron does not migrate to the interior of the catalyst but exists in the depth of 1~5 μm [32,33]. Because the type of metal deposition dictates the activity and behavior of the metal, which may have an influence on the physical and chemical properties of the catalyst, the deposition characteristics are important to study and quantify [34].

As shown in Figure 3, all catalysts were line scanned from center to edge. The distribution of iron on E-cat gradually decreased from the outside to the inside in Figure 4a. The iron mainly deposited on the outer surface of E-cat. In Figure 4b,c, the iron presented a uniform distribution from the center to the edge. During the process of the Mitchell impregnation, as the diameters of the iron chloride and iron naphthenate were smaller than that of the catalyst's pores, two kinds of iron-contaminated solution diffused into the catalyst pore at one time to form an equal-volume state. After calcination at a high temperature, the iron was evenly distributed in the catalyst. When using the Mitchell impregnation method to simulate iron contamination, due to the characteristics of the method, it is impossible to simulate the real reaction and regeneration environment of iron in an industrial plant. Therefore, iron was evenly distributed in the catalyst obtained using the Mitchell impregnation method.

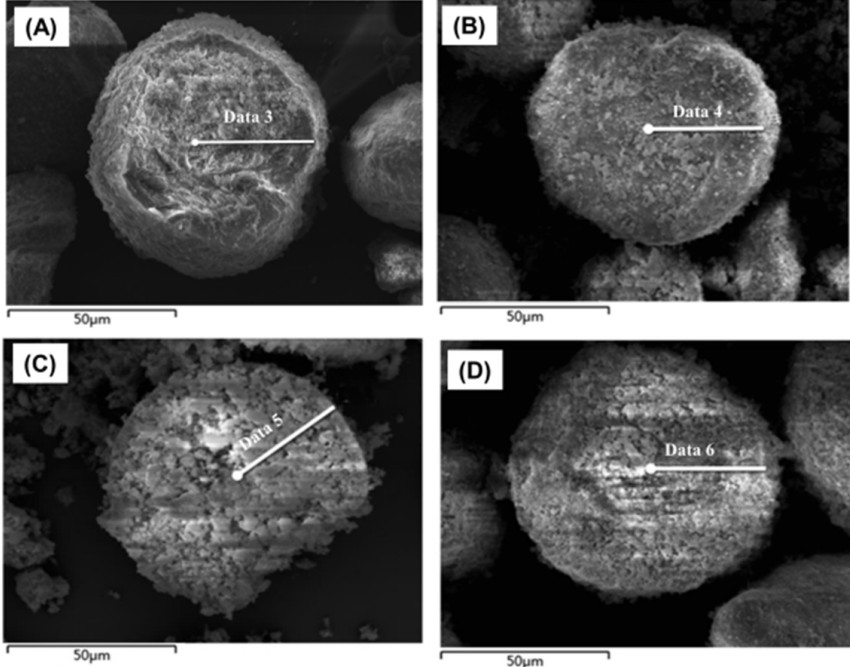

**Figure 3.** Line scan across a particle: (**A**) E-cat, (**B**) CAT-MM1, (**C**) CAT-MM2, and (**D**) CAT-CD.

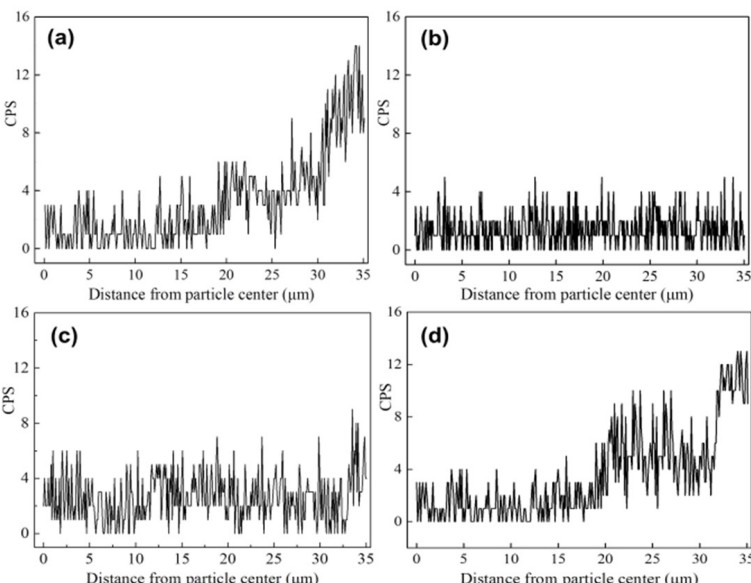

**Figure 4.** Iron distribution in a particle: (**a**) E-cat, (**b**) CAT-MM1, (**c**) CAT-MM2, and (**d**) CAT-CD. CPS: Counts Per Second.

The iron content in the depth of 1~5 µm from the edge on CAT-CD was higher. When moving closer to the center of the catalyst, the iron content decreased and tended to become balanced (Figure 4d). The distribution of iron on CAT-CD was similar to that of E-cat. The CD method used multiple cycles of reaction and regeneration steps to realize heavy metal deposition on the catalyst's surface. In a single cycle, 10 g of vacuum gas oil (VGO; Fe on VGO = 860 µg/g) was pumped into the quartz reactor to react with 150 g of the catalyst. Iron compounds were deposited on the catalyst surface after the cracking step. During the regeneration step, the iron compounds formed a eutectic with silica, alumina, and alkali metal compounds and released a lot of heat to melt the silica-alumina matrix layer to block and cover the pores of the catalyst. With an increase in the number of cycles, the pore closing became more serious. It was difficult for the newly deposited iron to penetrate the catalyst, which could only react and exist in the range of a few microns on the catalyst's surface. Therefore, the distribution characteristics of iron on an industrial equilibrium catalyst can be simulated by using the cyclic deactivation method.

### 2.4. Specific Surface Area

The trend of the zeolite, matrix, and total surface areas measured using nitrogen adsorption for E-cat, the Mitchell impregnated catalysts (with different iron precursors), and the cyclic-deactivated catalysts are shown in Figure 5.

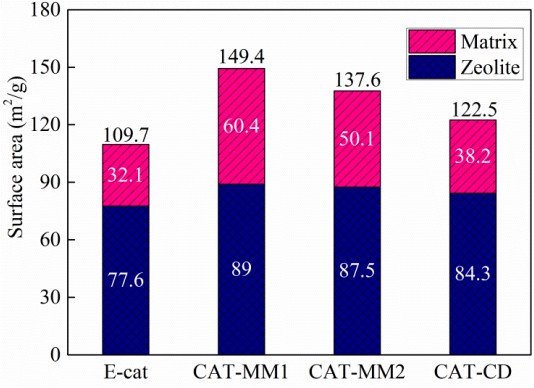

**Figure 5.** Zeolite, matrix, and total surface areas of the different catalysts.

Among the three deactivation methods, the total surface area of CAT-CD was the closest to that of E-cat. The zeolite and matrix surface areas were also studied. The zeolite surface was not significantly affected by different deactivation methods, while the matrix surface area changed greatly. Because of the uniform distribution of iron on CAT-MM1 and CAT-MM2, the pore closing effect had no serious effect on the matrix surface area. As iron is mainly deposited on the outer surface of CAT-CD and forms a layer that is a few microns thick, its matrix surface area was the lowest among the three methods. The reduction in the total surface area caused by the iron contamination was mainly due to the influence of the matrix. Therefore, an iron resistance catalyst may be designed from the perspective of improving the ability of the matrix to accommodate iron.

### 2.5. Iron Effects on the Catalyst Product Yields

The product yields of catalysts contaminated using the Mitchell impregnation and cyclic deactivation methods were measured using an ACE unit. The product yield curves were obtained by changing the catalyst-to-oil ratio (C/O) [35]. The conversion curves as a function of the catalyst-to-oil ratio for different catalysts are shown in Figure 6.

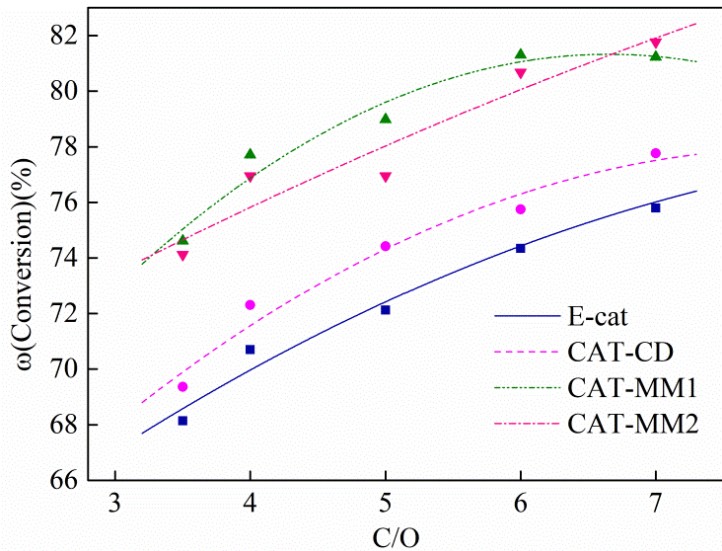

**Figure 6.** Conversion as a function of the catalyst-to-oil (C/O) ratio for different catalysts.

The catalysts' conversion increased with the increase of the C/O ratio. The adjustment of the C/O ratio could be realized by changing the catalysts' addition amounts under a fixed feed injection quantity. An increased C/O ratio meant that more catalysts were added and more active centers were able to function in the reaction, which eventually led to the increase in conversion.

The average conversion of the catalyst that was deactivated using the CD method was lower than that using the Mitchell impregnation method. The conversion values and trends of CAT-CD were also close to those of E-cat. During the process of iron contamination using the Mitchell impregnation method, the effect of pore closing on the catalyst was not serious, which led to the highest conversions for CAT-MM1 and CAT-MM2. A layer that was a few microns thick covered the surface of CAT-CD, which prevented the accessibility of the active sites of the catalyst may have resulted in an at least 4.84% decrease compared to CAT-MM1 and CAT-MM2.

The hydrogen factor curves of different catalysts are shown in Figure 7. The hydrogen factor is the mole ratio of hydrogen to methane in cracking products, which is an indicator of dehydrogenation reactions and can reflect the dehydrogenation ability of a catalyst [36].

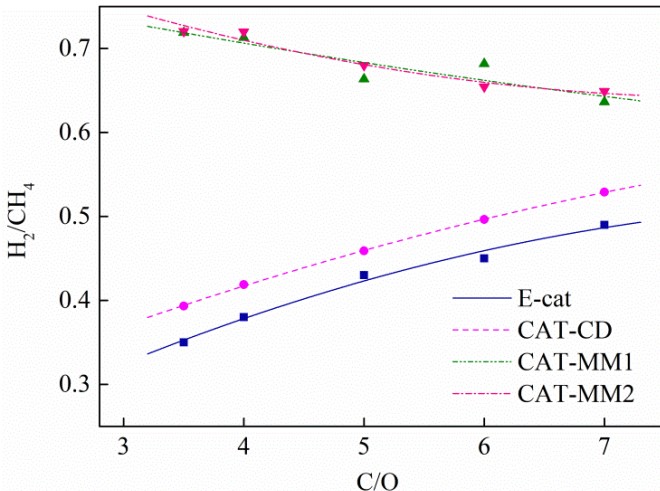

**Figure 7.** Hydrogen factor ($H_2/CH_4$ ratio) as a function of the catalyst-to-oil ratio for different catalysts.

The dehydrogenation ability of E-cat was the weakest, as part of the iron was passivated after a long period of operation in the FCC unit. Both CAT-MM1 and CAT-MM2 had the strongest dehydrogenation ability and their hydrogen factor values were close. Iron penetrated through the inner channels of CAT-MM1 and CAT-MM2. Therefore, it had a stronger dehydrogenation capacity under the same contamination content. The hydrogen factor curves were in the highest position. During the process of cyclic deactivation, part of the iron may have gathered in the nodules and had a weak dehydrogenation capacity. Therefore, the dehydrogenation capacity of CAT-CD was lower than for CAT-MM1 and CAT-MM2 under the same contamination content. CAT-CD could truly simulate the dehydrogenation capacity of an industrial iron-contaminated equilibrium catalyst.

The bottoms curves of the different catalysts are shown in Figure 8. According to the positions of the different curves, the bottoms yield of the catalyst that was deactivated using the Mitchell impregnation method was lower than that of E-cat. The matrix surface area of CAT-MM1 and CAT-MM2 was less affected. The feed molecules and products could move relatively freely in the catalyst channel. Therefore, the average bottoms yields of CAT-MM1 and CAT-MM2 were 4.4% lower than that of E-cat. The pore structure of CAT-CD was seriously affected by iron nodules such that its matrix surface area greatly decreased. It was difficult for the feed molecules to diffuse into the catalyst and contact the active center for cracking. As a result, the bottoms yield of CAT-CD greatly increased. Its bottoms curve was similar to that of E-cat.

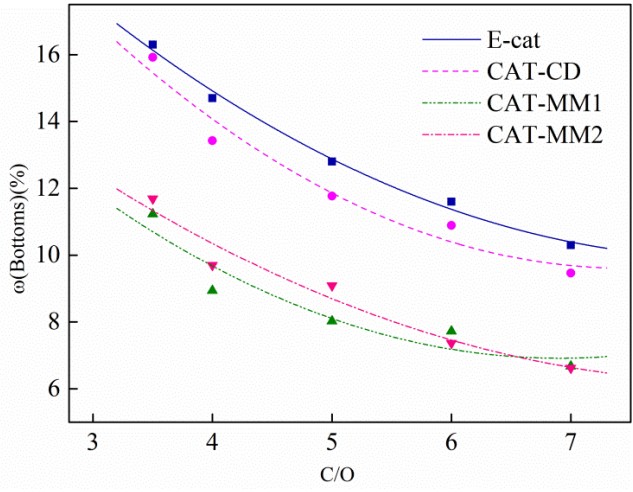

**Figure 8.** Bottoms as a function of the catalyst-to-oil ratio for different catalysts.

The products' selectivities toward the catalysts that were deactivated using different methods are indicated in Table 2 at the same 74 wt.% conversion.

**Table 2.** Product yields at 74 wt.% conversion.

| ω(Yields) (%) | E-Cat | CAT-CD | CAT-MM1 | CAT-MM2 |
|---|---|---|---|---|
| Coke | 9.18 | 9.33 | 10.29 | 9.87 |
| Dry gas | 3.28 | 3.22 | 3.24 | 3.06 |
| $H_2$ | 0.46 | 0.51 | 0.69 | 0.67 |
| Gasoline | 44.56 | 44.87 | 43.00 | 43.19 |
| LCO | 14.06 | 13.89 | 14.21 | 14.12 |
| Bottoms | 11.94 | 12.11 | 11.79 | 11.88 |
| LPG | 16.88 | 16.58 | 17.47 | 17.88 |

LCO: Light Cyclic Oil; LPG: Liquefied Petroleum Gas.

The coke and hydrogen selectivities of E-cat and CAT-CD were similar and were smaller than those of CAT-MM1 and CAT-MM2. This indicates that the iron on CAT-MM1 and CAT-MM2 had stronger dehydrogenation activity under the same amount of iron contamination, which produced more coke and hydrogen. Part of the iron in E-cat and CAT-CD was embedded and hence lost dehydrogenation activity, resulting in weak dehydrogenation activity. The bottoms selectivities of E-cat and CAT-CD were higher than those of CAT-MM1 and CAT-MM2, which indicates that the impregnated catalyst had a better cracking capacity for heavy oil. The product selectivity of CAT-CD was closer to that of E-cat.

### 2.6. $NH_3$-TPD Analysis of Different Catalysts

Figure 9 shows the $NH_3$-TPD (Temperature Programmed Desorption) profiles of E-cat and the catalysts prepared using different methods. The $NH_3$-TPD profiles demonstrate that the peak area of the weak and medium-strong acids of CAT-CD was smaller than those of CAT-MM1 and CAT-MM2. A pore-closing effect made it harder for the $NH_3$ molecules to get through the pores, which meant that the number of acid sites in CAT-CD was fewer. The number of acid sites was equal to the number of active sites. The number of active sites decreased in the following order: CAT-MM1 > CAT-MM2 > E-cat > CAT-CD. This was consistent with the change in the catalyst conversion and product distribution discussed before.

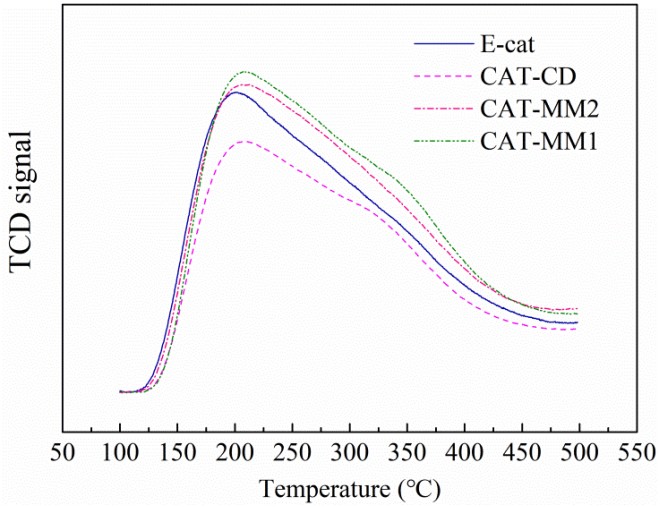

**Figure 9.** $NH_3$-TCD (Thermal Conductivity Detector) profiles of the different catalysts.

### 2.7. SEM-EDS Analysis of the Iron Nodule Formation Mechanism

The element content on the raised and valley areas of the catalyst deactivated using the cyclic deactivation method was observed in area scan mode using scanning electron microscopy (SEM) in combination with X-ray energy dispersive spectroscopy (EDS).

The distribution of silicon, aluminum, iron, calcium, and sodium in the selected area was analyzed using area scan mode. Six areas were selected as shown in Figure 10 for area scan analyze. The results are shown in Table 3.

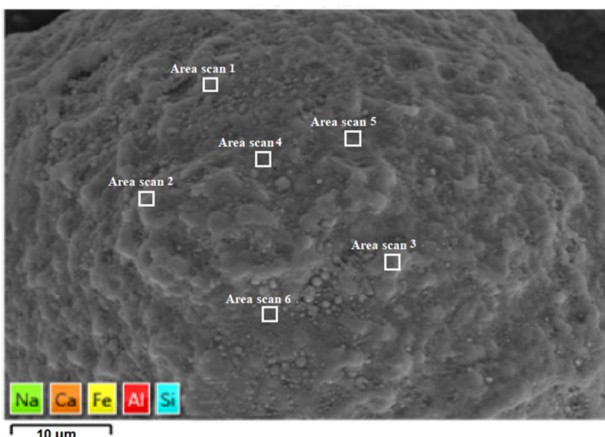

**Figure 10.** Area scan of a particle of CAT-CD.

**Table 3.** SEM-EDS analysis results of the element contents on CAT-CD.

| Location | Scanning Area | Si | Al | Fe | Na | Ca |
|---|---|---|---|---|---|---|
| Raised area on surface | Area scan 1 (%) | 34.16 | 48.89 | 15.18 | 1.77 | 0 |
| | Area scan 2 (%) | 33.55 | 43.62 | 18.29 | 4.53 | 0 |
| | Area scan 3 (%) | 38.91 | 43.42 | 16.36 | 1.31 | 0 |
| Valley area on surface | Area scan 4 (%) | 45.94 | 40.94 | 11.86 | 1.26 | 0 |
| | Area scan 5 (%) | 43.91 | 42.69 | 11.59 | 1.81 | 0 |
| | Area scan 6 (%) | 43.60 | 41.35 | 11.41 | 3.64 | 0 |

The content of aluminum was higher than that of silicon in the raised area, and it was the opposite in the valley area. The initial melting temperatures of the eutectic that consisted of $Na_2O$, $Al_2O_3$, and $SiO_2$ were higher than 770 °C. It was difficult for $Fe_2O_3$ to combine with $Al_2O_3$, $Na_2O$, and $SiO_2$ to form low-melting-temperature phases in areas that were rich in $Al_2O_3$; therefore, the structure of this area was preserved.

However, in the areas with a relatively high content of $SiO_2$, it was easy for the $Fe_2O_3$ to form a eutectic with $Na_2O$ and $SiO_2$, whose initial melting temperatures were lower than 500 °C [14]. During the regeneration process, the $SiO_2$ enrichment zone further intensified the melting and sintering, resulting in a regional collapse and local depression. The retention of the surface structure in the $Al_2O_3$-rich areas and the collapse of the surface structure in the $SiO_2$-rich areas resulted in a nodule morphology on the highly iron-contaminated catalyst surface.

The formation of iron nodules was the main reason for the various performances of the catalysts contaminated by iron. During the process of the Mitchell impregnation, the iron-contaminated solution filled the internal channels of the catalyst to form an equal-volume state. After the high-temperature calcining treatment, the iron was evenly distributed in the catalyst and there was no condition that allowed for the formation of iron nodules [37]. Therefore, iron nodules could not be observed on the catalyst's surface and the deactivated catalyst could not simulate an iron-contaminated equilibrium catalyst.

Iron deposition on the surface of the cyclic-deactivated catalyst was realized using long-time repeated cracking–regeneration cycles. The iron deposition in a single cycle was

very low. After the cracking step, iron oxide combined with silicon oxide, aluminum oxide, and alkali metal oxides to form low-melting-temperature phases before it diffused inside the catalyst. The formation conditions of iron nodules in industrial plants could be simulated by using the cyclic deactivation method. The CD method is an effective and accurate method for simulating iron contamination in a laboratory.

### 3. Materials and Methods

*3.1. Materials*

In this research, a commercial catalyst (CAT-BASE) without iron contamination, which was used to simulate iron deactivation, was collected from the industrial plant of the Dalian West Pacific Petrochemical Co., Ltd. (WEPEC, Dalian, China). In order to simulate the actual state of an iron-contaminated equilibrium catalyst in an FCC unit, the target amount of iron deposition and appropriate conditions were selected to carry out Mitchell impregnation and the cyclic deactivation method. The equilibrium catalyst (E-cat) with a high level of iron contamination was also collected from WEPEC.

*3.2. Methods*

3.2.1. Mitchell Impregnation

Iron chloride and iron naphthenate were used as contamination species to carry out this method. Iron chloride was added to deionized water (Fe on solution = 14,400 µg/g) and iron naphthenate was added to a toluene solution (Fe on solution = 12,500 µg/g). The catalysts (150 g) were impregnated with 120 g of iron chloride solution and 150 g of iron naphthenate solution, respectively, dried for 2 h at 150 °C, and then calcined for 6 h at 600 °C before use. During the drying process, the catalyst was stirred every 20 min to ensure that it was evenly heated and would not agglomerate. The catalysts that were impregnated with iron chloride and iron naphthenate were labeled as CAT-MM1 and CAT-MM2, respectively.

3.2.2. Cyclic Deactivation

The deactivated catalyst was treated using 200 cycles of cracking, stripping, regeneration, and cooling in the MCD (Multi Cyclic Deactivation) unit produced by 360° KAS Company (Amsterdam, The Netherlands). The cracking step was performed at a reaction temperature of 530 °C and the regeneration step was performed at 780 °C. During the cracking step, 10 g of VGO spiked with iron naphthenate (Fe on VGO = 860 µg/g) was pumped into the quartz reactor to react with 150 g of the catalyst at the reaction temperature. Finally, the aged catalyst was marked as CAT-CD.

3.2.3. Catalyst Characterization

The metal content was determined using a ZSX PrimusIIX-ray fluorescence instrument produced by the Japan Science and Technology Company (Tokyo, Japan). The X-ray tube voltage was 50 kV, the tube current was 50 mA, and the diaphragm aperture was 20 mm.

The catalyst phase structure was determined using a D/max-3C X-ray diffractometer produced by the Japan Science and Technology Company (Tokyo) with the following settings: Cu target, Kα radiation, tube voltage: 40 kV, tube current: 20 mA, scanning range: 5 to 50°, scanning speed: 4°/min.

$NH_3$-TPD was performed on a Micromeritics AUTOCHEM II 2920 produced by the Micromeritics Instruments Corporation (Norcross, GA, USA). 0.1 g of sample was purified with He gas flow at room temperature for 1 h. $NH_3$ adsorption was performed by adjusting temperature with two-stage method. The desorption curve was obtained after the adsorption is stable, and the temperature was raised to 500 °C within 50 min.

The specific surface area and pore volume of the catalyst were measured using a Micromeritics ASAP 3000 automatic physical adsorption instrument produced by the Micromeritics Instruments Corporation (Norcross, GA, USA). The surface area was deduced from the adsorption isotherms using the BET (Brunauer, Emmett and Teller) equation.

The catalyst surface morphology was observed using an ULTRA PLUS thermal field emission scanning electron microscope produced by Zeiss Optical Instruments (Oberkochen, Germany). Secondary electron resolution: 1.0 nm (15 kV) and 1.9 nm (1 kV); electron gun: LaB6 thermal field emission electron gun; acceleration voltage: 0.1~30 kV, magnification: ×12~×10000.

The iron distribution on the catalysts' cross-section was analyzed using line scan mode during SEM in combination with an X-ray energy dispersive spectroscope produced by Oxford Instruments (Abingdon, UK). Resolution: better than 127 eV; The MnK$\alpha$ peak-to-back ratio: 20000:1; stability: 1000~100,000 cps, spectral peak drift <1 eV, resolution change <1 eV. All four catalysts were measured from the center to the edge. In order to ensure the accuracy of the measurement, a total of 350 points were measured by using a spot analysis technique over a length of 35 μm to make up the results of the line scan.

### 3.2.4. Catalyst Cracking Activity and Selectivity

The cracking performance of different catalysts was tested in an Advanced Cracking Evaluation unit developed by the KTI Technology Company (Huston, TX, USA). The flowchart of the ACE unit is shown in Figure 11. The reaction conditions were the following: The amount of catalyst added was 9 g. The reaction and regeneration temperatures were 530 °C and 715 °C, respectively. The variation of the catalyst-to-oil ratio (C/O = 3.5, 4.0, 5.0, 6.0, 7.0) was performed by varying the catalyst amount. Gaseous products were analyzed using an GC-3000 on-line chromatograph produced by INFICON Company (East Syracuse, NY, USA) according to the UOP method 539. GC-3000 uses four chromatographic modules for detection, the work temperature is 0 °C~50 °C. The carrier gases used are helium, hydrogen, nitrogen and argon. The simulated distillation of liquid products was carried out using a 7890B chromatograph produced by Agilent Technologies Inc. (Santa Clara, CA, USA) according to the SH/T 0558 procedure. The working environmental temperature of Agilent 7890B is 15 °C~35. Retention time repeatability <0.0008 min. Carrier and makeup gas settings selectable for helium, hydrogen, nitrogen and argon/methane. Coke deposited on the catalyst was quantified using a $CO_2$ analyzer produced by Servomex Group Co., Ltd (Sussex, UK). The detection range of $CO_2$ analyzer is 0~20%. When the detection value is lower than 0.4%, the regeneration step is considered to be completed. Conversion and yields of dry gas ($H_2$ + C1 + C2), LPG (Liquefied Petroleum Gas) (C3 + C4), gasoline (C5 < bp < 221 °C), LCO(Light Cyclic Oil) (221 °C < bp < 343 °C), HCO (Heavy Cyclic Oil)(bp > 343 °C), and coke were calculated.

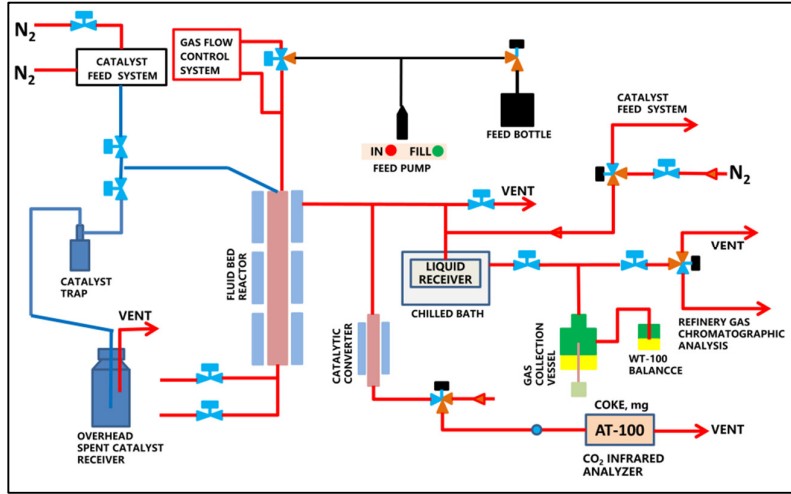

**Figure 11.** The flowchart of the Advanced Cracking Evaluation (ACE) unit.

## 4. Conclusions

Iron nodules are the main factor that leads to the poor performance of iron-contaminated catalysts. The retention of the surface structure in the alumina-rich areas and the collapse of the surface structure in the silica-rich areas resulted in a continuous nodule morphology on a highly iron-contaminated equilibrium catalyst.

Iron was evenly distributed in the catalyst prepared using the Mitchell impregnation method, and no obvious iron nodules or boundary were found on the surface of the catalyst. The surface area was less affected by iron contamination. Iron deposited on the catalyst resulted in a small reduction in the conversion and it had little effect on the bottoms selectivity, but it led to a strong dehydrogenation capacity. The catalytic performance of the impregnated catalyst was quite different from that of the iron-contaminated equilibrium catalyst.

The iron was mainly present in the depth of 1~5 μm from the edge on the catalyst prepared using the cycle deactivation method. As iron nodules formed on the surface of the cyclic-deactivated catalyst, pore closing limited the feed molecules' diffusion inside the particles and the products' diffusion outward, resulting in a significant decrease in conversion, extremely poor bottoms selectivity, and a small increase in dehydrogenation capacity.

We established a laboratory simulation method that can accurately simulate the industrial iron-contaminated equilibrium catalyst. By using this method, we can recognize the real mechanism of iron contamination, understand the main factors that affect the physical and chemical properties of an iron-contaminated catalyst, and predict the actual reaction performance and product distribution of a catalyst prepared in a laboratory. This can effectively reduce the time and economic costs caused by the failure of laboratory simulations.

In our next research work, we can rely on the accurate CD method to develop iron tolerance technology. On the other hand, we can also investigate the combined effect of iron and other heavy metals, such as nickel and vanadium, on the catalyst to further improve the iron deactivation simulation method.

**Author Contributions:** Y.L. and T.L. conceived and designed the experiment; Y.L. performed the experiments and analyzed the data. All the authors listed have made contributions to drafting and writing the manuscript. All authors have read and agreed to the published version of the manuscript.

**Funding:** This research was funded by the Ministry of Science and Technology Management of PetroChina (2019D-5006-20).

**Institutional Review Board Statement:** Not applicable.

**Informed Consent Statement:** Not applicable.

**Data Availability Statement:** Data available in a publicly accessible repository.

**Acknowledgments:** We acknowledge the financial support from the Development of New Catalysts with Low Emission and High Gasoline Yield project, the Ministry of Science and Technology Management of PetroChina (2019D-5006-20). We thank L. Zhang for kind assistance in SEM-EDS experiments, and X.Q. Sun for her kind helps in NH$_3$-TPD analysis.

**Conflicts of Interest:** The authors declare no conflict of interest.

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
