# Peer review of "A Comparison of Laboratory Simulation Methods of Iron Contamination for FCC Catalysts"

_catalysts, doi:10.3390/catal11010104_

Round 1

Reviewer 1 Report

COMMENTS TO MANUSCRIPT NUMBER catalysts-1053582.

Title: A Comparison of Laboratory Simulation Method of Iron Contamination for FCC Catalysts

I recommend the manuscript to be ACEPTED AFTER MAJOR REVISION.

This paper studies two different methods of simulating iron contamination for FCC catalysts in laboratory scale. I think that the topic of the paper is quite interesting but I have found it quite poor regarding to the experimental and definitions. I have found lack of information.

2.1section is quite poor. Properties of the CAT-BASE are missing. Properties of the prepared catalysts are missing. Which methods have you used for metal loading?

Which catalyst is E-cat? It is not clear for me the nomenclature used.

Could you insert in the manuscript the scheme of ACE unit?

Which is the  definition used for conversion?

Reviewer 2 Report

Catalysts 1053582

A Comparison of Laboratory Simulation Method of  Iron Contamination for FCC Catalysts

Authors:  Yitao Liao, Tao Liu, Huihui Zhao, Xionghou Gao

The authors study the Fe contamination on FCC catalysts. The catalysts were characterized with various techniques and evaluated for their activity. It was found that Fe decreases the conversion and bottoms selectivity, while it increases dehydrogenation capacity. The two different methods result in different distribution of Fe deposits and the cyclic deactivation method is more realistic as a simulation method

The study deals with a well-known problem, the deactivation of catalysts by metal deposits although Ni and V are the most common and better studied metals.

The manuscript is well written and easy to follow. The results are good and the article can be accepted for publication after revision.

Comments

A small paragraph about the bibliographic data for Fe contamination should be added in Introduction part

 A definition of E-Cat can be given in 2.1 or 2.2 section

The iron distribution of the two Mitchell impregnation it seems strange to me. I can understand the uniform distribution but the total Fe seems less than the total Fe on the used catalysts and CD catalyst. Also, it may be a difference in the distribution of MM1 and MM2 catalyst. The second seems like an egg white distribution with a peak at 15-20 μm. The difference can be due to the different precursor of the FE salt and the absence of ion pair interactions. Please comment the above especially the low value of the Fe in the uniform distribution catalysts (MM1 and MM2) and how the impregnation can lead to these different see for example Applied Catalysis A: General 399, 211-220. Finally, the profile of Fe in CD catalyst is also complicated. There are some small differences with the E-Cat but I believe that they can be attributed to the different concentration of Fe in the feed.

Lines 148-150 Please rephrase

Can the authors provide selectivity data under the same conversion? This is the most safe way to check differences in selectivity since the local concentration of the reactants can alter significant the selectivity, and thus, the performance of the catalyst.

The author can discuss possible changes in the acidity of the catalysts due to Fe deposition.

Can the authors provide TEM results to support their findings about FE deposition?

The mechanism of dehydrogenation can be addressed and discussed with respect to the Fe in order to support the Fe action

Reviewer 3 Report

I like this manuscript. It describes an original and interesting work. As such, it has the potential to be published in Catalysts. However, I have some comments that the authors should implement into the revised manuscript before publication.

1) FCC - Please, explain this acronym (also in the title and in the abstract).

2) Introduction - Please, avoid cumulative references (see, e.g., [13-18] and [21-26]) and explain the main contribution of each cited paper.

3) Introduction - The connection between the aim and the literature gaps has to be better described, thus giving more strength to the reason behind this work.

4) Conclusions - The practical impact of the results obtained in this work should be better highlighted.

5) Conclusions - The authors should also give an outlook on future research.

I’m willing to review the revised manuscript.

Round 2

Reviewer 1 Report

I do agree with the changes made by the authors. 

Reviewer 2 Report

The authors revised their manuscript following the reviewer's comments. It can be accepted for publication.